# A Scoping Review to Assess Adherence to and Clinical Outcomes of Wearable Devices in the Cancer Population

**DOI:** 10.3390/cancers14184437

**Published:** 2022-09-13

**Authors:** Yaoru Huang, Umashankar Upadhyay, Eshita Dhar, Li-Jen Kuo, Shabbir Syed-Abdul

**Affiliations:** 1Department of Radiation Oncology, Taipei Medical University Hospital, Taipei 110, Taiwan; 2Graduate Institute of Biomedical Materials and Tissue Engineering, College of Biomedical Engineering, Taipei Medical University, Taipei 110, Taiwan; 3Graduate Institute of Biomedical Informatics, College of Medical Sciences and Technology, Taipei Medical University, Taipei 106, Taiwan; 4International Center for Health Information Technology, College of Medical Science and Technology, Taipei Medical University, Taipei 106, Taiwan; 5Division of Colorectal Surgery, Department of Surgery, Taipei Medical University Hospital, Taipei 110, Taiwan; 6Department of Surgery, School of Medicine, College of Medicine, Taipei Medical University, Taipei 110, Taiwan; 7School of Gerontology and Long-Term Care, College of Nursing, Taipei Medical University, Taipei 110, Taiwan

**Keywords:** wearable devices, health monitoring, cancer, eHealth

## Abstract

**Simple Summary:**

The use of wearable devices in clinical care is gaining popularity among cancer patients. The COVID-19 pandemic highlighted the value of wearable devices for monitoring health. Wearable devices are used to record and monitor real-time data like physical activity, sleep metrics, and heart rate variables. The use of wearable devices can directly impact clinical decision-making. There are few pieces of evidence that prove that wearable could improve the quality of patient care while reducing the cost of care, such as remote health monitoring. The generated big data by the wearable device is both a challenge and an opportunity. Researchers can apply artificial intelligence and machine learning techniques to improve wearable devices and their usage among cancer patients. In this scoping review, we assessed the adherence to clinical outcomes of wrist-worn wearable devices in the cancer population.

**Abstract:**

The use of wearable devices (WDs) in healthcare monitoring and management has attracted increasing attention. A major problem is patients’ adherence and acceptance of WDs given that they are already experiencing a disease burden and treatment side effects. This scoping review explored the use of wrist-worn devices in the cancer population, with a special focus on adherence and clinical outcomes. Relevant articles focusing on the use of WDs in cancer care management were retrieved from PubMed, Scopus, and Embase from 1 January 2017 to 3 March 2022. Studies were independently screened and relevant information was extracted. We identified 752 studies, of which 38 met our inclusion criteria. Studies focused on mixed, breast, colorectal, lung, gastric, urothelial, skin, liver, and blood cancers. Adherence to WDs varied from 60% to 100%. The highest adherence was reported in the 12-week studies. Most studies focused on physical activity, sleep analysis, and heart vital signs. Of the 10 studies that described patient-reported outcomes using questionnaires and personal interviews, 8 indicated a positive correlation between the patient-reported and wearable outcomes. The definitions of the outcome measures and adherence varied across the studies. A better understanding of the intervention standards in terms of the clinical outcomes could improve adherence to wearables.

## 1. Introduction

Considerable advancements have been made in the fields of biosensors and artificial intelligence, particularly in terms of their use for the detection and management of chronic illnesses [1]. Wearable devices (WDs) are electronic devices that can be easily worn on the human body to capture real-time healthcare data using receptors and transducers attached to the device [2]. According to Deloitte Global’s prediction analysis, 440 million individuals will be using WDs by the end of 2024, with more healthcare providers recommending their use and more consumers becoming more comfortable with using them in their daily lives [3]. WDs include any device that can be worn on the human body including wristwatches, glasses, chest straps, rings, and prosthetic sockets [4]. WDs are mainly used to track individuals’ daily activities, including sleep quality, physical activity, and heart rate [4,5]. The evolution of smart WDs is being accelerated by improved sensors, artificial intelligence, and advanced machine-learning algorithm–based technologies [6]. For example, few watches that are currently available have optical sensors that can collect real-time data on blood physiology and blood pressure through photoplethysmography [7].

Several studies have reported the use of wearables in the healthcare field with promising outcomes [8]. WDs have revolutionized the healthcare system and have reduced the load of hospitals by providing reliable information in a timely manner [8,9]. Various WDs with a wide range of types and employability are available for data collection [10]. The rich information collected by WDs can assist healthcare professionals in tracking a person’s health status (sleep quality, healthy posture, cognitive decline, and even early warning signs of infection and inflammation) [8,9,10]. Chronic health conditions result in a high financial and emotional burden on patients and their families [11]. In response to COVID-19, many healthcare professionals reengineered their pathways to promote “care in place,” which allows patients to track their health and participate in the self-care system [12].

Wearables in oncology may provide new, vital information on a patient’s health status (heart rate, blood pressure, activity level, sleep quality, and behavioral activity), which can improve cancer care management [10]. 

Cancer is the second leading cause of death, and treatment is costly; however, using wearable devices can be cost-effective as they require personalized and flexible patient treatment plans [13,14]. Patients can also track their own data while continuing their normal routines while their data are being transferred to the clinic; this approach in oncology is a work-smarter approach and, more importantly, provides a general improvement in the quality of life [15,16]. Few studies provided evidence regarding the effectiveness of WDs in improving the treatment outcomes of patients with cancer [17].

To understand the potential use of WDs in cancer management, the effect of an intervention on or the role of WDs in clinical outcomes should be investigated [10]. Evaluating patients’ outcomes and their adherence to WDs, as well as defining the criteria and valid data, can aid in introducing WDs into clinical practice [18]. Although a consensus and guidelines for designing and reporting trials that use wearables as a component of the intervention are lacking, this area of research is gaining attention [19] because WDs play a major role in healthcare management [8]. This scoping review examined the effectiveness of WDs in assisting patients in their care, particularly in cancer treatment. In addition, this review evaluated the effectiveness and feasibility of this type of intervention. The goal of this review is to explore the use of WDs in patients with different types of cancers given the increasing number of WD users.

## 2. Methods

### 2.1. Search Strategy

The search strategy was conducted in accordance with the Preferred Reporting Items for Systematic Reviews and Meta-Analysis guidelines. A comprehensive literature search was conducted to identify scientific studies that analyzed WD-based interventions targeting patients with cancer and cancer survivors. We searched for studies published between 1 January 2017 and 7 March 2022, in the electronic databases PubMed, Scopus, and Embase. The search for studies was conducted on 9 March 2022. The search was conducted using MeSH (medical subject headings) such as wearables and cancer/devices and cancer/telemonitoring and cancer. The search was limited to articles published in English. The data collected were reviewed by the two authors of this study. If any reviewer considered an article to be potentially significant, the full text of the article was retrieved. In the case of a disagreement on a particular article, a third reviewer chose the article based on the exclusion and inclusion criteria.

### 2.2. Criteria for the Inclusion of Studies

We included original research articles that were published in English and met the following criteria: (1) included cancer survivors and patients with cancer undergoing treatment; (2) focused on preventive care or health monitoring (physical activity, behavioral activity, and quality of life); (3) used a smartwatch or other types of wrist-worn WDs for assessing health; and (4) were designed as a randomized controlled trial (RCT), prospective clinical trial, quasi-experimental study, feasibility study, observational study, or pilot study.

We excluded studies that (1) focused on the prediction of cancer; (2) used telecommunication technologies, such as websites, telephones, and mobile applications alone; (3) did not involve the use of wrist-worn wearables or focused on some other health problems; (4) did not include the intervention as the primary focus; and (5) were review articles, trial protocols, trial registrations, conference papers, book chapters, notes, brief reports, letters, editorials, or case studies or were published in a language other than English.

### 2.3. Study Selection and Data Extraction

We initially screened the titles and abstracts of articles that met the inclusion criteria and could not be rejected with certainty. The references of the studies that met the inclusion criteria were manually searched to identify relevant research articles. Three researchers independently reviewed the full texts of the articles and extracted the following data from each included article: characteristics of the study (country of origin, sample size, authors, study design, study purpose, and publication year); patient characteristics (mean age, sex percentage, cancer type, and health status); intervention characteristics (total study duration including the follow-up period, intervention duration, and type of wearable and tools used); and study focus (behavioral health monitoring and preventive care). For this scoping review, data on adherence to wearables were extracted according to the different criteria mentioned in each article. Adherence was measured in terms of the percentage of valid wear time among patients or the percentage of total patients who completed the trial and the percentage of total evaluable days [20]. Furthermore, we categorized the studies based on the methods and outcomes (Figure 1).

### 2.4. Adherence Analysis

Adherence analysis results were extracted from the studies and analyzed according to the different criteria mentioned in the selected articles. The majority of included studies evaluated adherence through the completeness of the data collection, that is, the percentage of recruited patients who completed the study. The remaining studies had specific criteria for the adherence evaluation that are mentioned in results section. We used SPSS for visualization purposes in order to graphically represent adherence to wearable devices by the duration of the study intervention.

### 2.5. Outcomes and Analysis

This review examined the effectiveness of WDs in health monitoring (symptom analysis/recovery assessment/physical activity, behavioral activity, quality of life, and preventive care) among patients with cancer undergoing treatment (chemotherapy, radiotherapy, or pre- and post-surgery treatment) and cancer survivors. The primary outcome was adherence to wearables according to the different criteria of each study. The secondary outcomes included the wearable, patient-reported, and clinical outcomes of the intervention. All data are presented in a descriptive manner.

### 2.6. Ethical Consideration 

This review did not require the approval of any national or institutional boards.

## 3. Results

### 3.1. Study Selection

A total of 752 articles were retrieved from Embase, Scopus, and PubMed. After the removal of the duplicates, 473 articles were evaluated. Finally, of the 473 articles, the full texts of 102 relevant articles were reviewed. At this stage, the articles were screened based on their study characteristics, including the type of wearables and their usage, and the primary focus of the study. The exclusion criteria are mentioned in the methodology and presented in Figure 1. Finally, 38 studies that met the inclusion criteria were included in this scoping review.

### 3.2. Study Characteristics

Among the 38 studies, 19 were conducted in North America (the United States and Canada), 6 in Europe (Ireland, Germany, the United Kingdom, the Netherlands, France, and Switzerland), 9 in Asia (Taiwan, Central China, India, Japan, and South Korea), and 4 in Oceania (Australia; Table 1).

Among the 38 studies, 12 were designed as RCTs, 7 as feasibility studies, 5 as observational studies, 4 as pilot studies, 3 as cohort studies, 2 as nonrandomized controlled trials, 2 as usability studies, 1 as a utility study, and 1 as a group and qualitative study (Table 1). The included studies had different time intervals for the intervention and follow-up durations. The minimum and maximum intervention durations were 1 and 52 weeks, respectively. All the studies had the same follow-up period of 12 weeks (Table 1).

### 3.3. Characteristics of Research Participants

Most of the studies included patients with different types of cancer, followed by those with breast, colorectal, lung, gastric, urothelial, skin, liver, and blood cancers. Of the 38 studies, 15 focused on cancer survivors and 23 on patients receiving chemotherapy or radiotherapy. The number of participants in the studies ranged from 8 to 555 and their mean age ranged from 17 to 73 years (Table 1).

### 3.4. Measurement Tools

In terms of measurement tools, the studies included in this scoping review used subjective self-reported questionnaires, WDs, and mobile applications to track the health status of the participants. Only four studies conducted personal interviews to analyze participants’ adherence to WDs and their study experiences [21,33,40,57]. Furthermore, 10 studies evaluated the correlation between the outcomes of WDs and questionnaires to validate the effectiveness of using WDs in patients with cancer [27,31,34,36,37,48,49,52,54,56]. Only two studies used an e-diary and a physical diary along with a WD to track the health status of the participants [31,36].

### 3.5. Major Study Focus

The studies included in this review mainly focused on physical activity, sleep quality, quality of life, unplanned healthcare encounters, moderate-to-vigorous physical activity (MVPA), sedentary behavior, and symptom burden. Only one study focused on preventive care management (Table 1).

### 3.6. Intervention Methods

The studies included in this review article employed different interventional approaches to improve the clinical outcomes of the participants. Most of the studies used a mobile application to provide the intervention [22,34,35,39,40,46,53,54,57]. Of the 38 studies, 3 provided the intervention by sending text messages [22,45,57], and 4 conducted a general group session or included a virtual support group [39,41,44]. Furthermore, 7 studies used in-app chat services; conducted behavioral counseling sessions, coaching programs, and group phone calls; and provided health education [21,22,23,24,30,38,42,46]. The remaining studies included a few other approaches (Table 1).

### 3.7. Interventional Outcomes

#### 3.7.1. Adherence

The adherence rate was calculated for both the follow-up and intervention periods by determining the average time of the intervention across the 38 studies. Of the 38 studies in this review, 26 examined adherence in terms of the completeness of the data collection, and the remaining 12 studies followed different criteria to examine adherence (Table 2). Of the 38 studies, 6 determined the acceptability of WDs by conducting qualitative personal interviews and evaluating the usage of the device on different days during the study [24,29,45,47,52,57]. Seven studies determined feasibility and four studies evaluated the retention of WDs according to the same criteria used for determining acceptability [4,20,21,23,24,39,41,42,45,47,52,55]. Only one study focused on adherence based on the completion of an exercise program [55].

The adherence during the intervention periods was visualized graphically using SPSS. Figure 2 shows the adherence rate to the exhibited intervention duration. All of the studies were divided into 11 segments based on the duration of each study’s intervention. The average adherence percentage was calculated for each segment. The intervention-based segments included studies with weeks 1 [48,49], 2 [27,55], 3 [50,52,57], 4 [36,37,53], 7 [28], 8 [32,33,34], 10 [24,30,41], 12 [20,21,23,25,26,38,39,40,42,43,44,45,46,47,54,56], 17 [29], 24 [22,31] and 43 [35]. In the graph, (1.84) indicates the studies grouped in this segment that had a one-week intervention with an average adherence of 84%. According to our analysis, 16 out of 38 studies used a 12-week intervention period with an average adherence of 86% (Figure 2) [20,21,23,25,26,38,39,40,42,43,44,45,46,47,54,56]. The follow-up periods in the reported studies were calculated by subtracting the intervention duration from the total study duration. There are only five studies with a 12-week follow-up period, accounting for 24 weeks of total study duration. These studies showed an average adherence rate of 91% [20,21,25,38,43]. Studies with an intervention period of 4 or 7 weeks also observed high adherence to WDs (i.e., 92% and 93%, respectively) [28,36,37,53] (Figure 2). There is one unique study that enrolled patients across four seasons of the year. This study showed a 100% adherence rate throughout the 52-week study duration [52]. The adherence outcomes are summarized in Table 2.

#### 3.7.2. Clinical Outcomes

Wearable technology can directly affect clinical decision making and improve the quality of patient care while reducing the cost of care including that of patient rehabilitation outside of hospitals. The studies in this review focused on physical activity, sleep quality, quality of life, unplanned healthcare encounters, MVPA, sedentary behavior, and symptom burden.

Most of the studies in this review evaluated the feasibility of and adherence to WDs. Of the 38 studies, only 10 determined the correlation between patient-reported and objective outcomes; these studies reported that the use of WDs significantly improved clinical outcomes [27,31,34,36,37,48,49,52,54,56]. In addition, other studies demonstrated significant improvements in clinical outcomes; however, these studies did not observe a correlation between patient-reported and clinical outcomes. Of the 38 studies, 12 included patients with different types of cancer, 6 included cancer survivors, and 11 included patients with breast cancer. Of the 11 studies, 5 focused on patients receiving chemotherapy or radiotherapy. Moreover, of the 38 studies, 5 included patients with colorectal cancer, 4 included patients with lung cancer, and 2 included patients with gastric cancer. Furthermore, 4 of the 38 studies included patients with urothelial carcinoma, skin cancer, blood cancer, and liver cancer. The clinical outcomes are summarized in Table 3.

## 4. Discussion

### 4.1. Summary and Findings

The use of WDs as gadgets for tracking daily activities, particularly physical activity, has become widespread [59]. WDs are also used for patient and disease management. To the best of our knowledge, this is the first scoping review to examine adherence to WDs in the cancer population. We performed a scoping review of articles evaluating the heterogenous use of wrist-worn devices in patients with cancer. By using our search strategy, we initially retrieved 752 studies, of which 38 were finally included in this scoping review. Most of the studies in the review included patients with mixed cancer types, followed by those with breast cancer. In addition, other studies included cancer survivors and patients with cancer receiving chemotherapy or radiotherapy. The included studies were designed as RCTs and nonrandomized controlled, observational, feasibility/pilot, cohort, and group studies. The sample sizes ranged from 8 to 555. All the studies included used either subjective questionnaires or mobile applications along with a WD. The intervention duration varied from 1 to 52 weeks. Furthermore, 89% (34/38) of the studies evaluated physical activity as the clinical outcome. The outcomes of using WDs varied among the studies based on the intervention program and the usage rate of WDs. Of the 38 studies, 10 compared WD outcomes and patient-reported outcomes (determined using subjective questionnaires) and examined whether the use of WDs improved clinical outcomes. Of the 10 studies, 8 reported that using WDs considerably improved clinical outcomes. The study designs and outcomes varied among the included studies.

### 4.2. eHealth Tools for Cancer Care

Other studies explored the use of eHealth tools involving patient self-reporting of medication and healthcare management and their effects on the health of users [60]. More high-quality studies are warranted before the standard implementation of eHealth tools. In oncology, patients are increasingly required to manage their own illnesses; thus, WDs can be a valuable tool in the management of cancer during therapy [61]. However, technical and clinical adherence to such devices are essential aspects that should be explored because they determine the usage rate of devices among patients [62]. In this scoping review, we evaluated the adherence to WDs that was reported in the included studies. The adherence rate can be calculated using various factors including wear time, the number of patients using the device, data collection while wearing the device, and the number of evaluable days [20]. A wide range of data were collected during the days on which the device was worn and no data were collected on the days on which patients missed wearing the device. Because of the variations in the data collection and adherence, the effectiveness of WDs for the health management of patients with cancer remains unclear. Whether patients wear the device when they feel comfortable based on the provided intervention should be evaluated. Thus, before designing WD-related studies or large interventional studies, we need to define the criteria and set a fixed wearable time to understand the adherence to devices.

### 4.3. Strength and Limitations

To the best of our knowledge, this scoping review is the first to determine adherence to specific wrist-worn devices. However, the choice of a WD and its outcomes are crucial. The selection of appropriate clinical variables for measurement is crucial based on the purpose of WD use when incorporated into patients’ daily routines. The outcomes must be based on the type of disease. We agree with other researchers that monitoring specific outcomes is crucial because unnecessary outcomes would not substantially affect patients’ treatments. Most of the studies investigated the effectiveness of WDs for improving physical activity and quality of life; however, this may be attributed to the increasing knowledge of fitness and increased physical training. Furthermore, the secondary objective of most of the studies was to examine the clinical outcomes of WDs in terms of healthcare management. Only 10 studies reported a positive correlation between the use of WDs and patient-reported outcomes.

Since each article used its own definition, it was not possible for us to compare the adherence data of all studies, which is one of the significant limitations of our review. Another limitation is that the included studies mainly involved patients with mixed cancer types, followed by those with breast cancer. Moreover, most of the studies were conducted in North America and Europe. Thus, the findings of this study might not be applicable to populations from other geographical areas. We were unable to conduct a meta-analysis or systematic review due to the numerous variations in the study design, clinical outcomes, and adherence definitions. As a result, we conducted a scoping review and presented a tabulated analysis of our studies.

## 5. Conclusions

This study reports that the definitions of the outcome measures and adherence varied across the studies. There was a limited consensus among the studies for the measured variables during treatment. Adherence to wearable devices was affected by the changes in the intervention or study design. A better understanding of the interventional period of wearable devices in terms of clinical outcomes is urgently needed. Studies using WDs and subjective questionnaires encouraged patient engagement for better cancer care management. Adherence to WDs varied from 60% to 100% depending on the intervention period. The highest adherence was reported in the 12-week studies. Most studies focused on physical activity, sleep analysis, and heart vital signs. Of the 10 studies that described patient-reported outcomes using questionnaires and personal interviews, 8 indicated a positive correlation between patient-reported and wearable outcomes. Furthermore, for a better understanding of adherence behavior, we need large intervention studies. This can provide us with a clear picture of the clinical outcomes of using wearable devices.

## Figures and Tables

**Figure 1 cancers-14-04437-f001:**
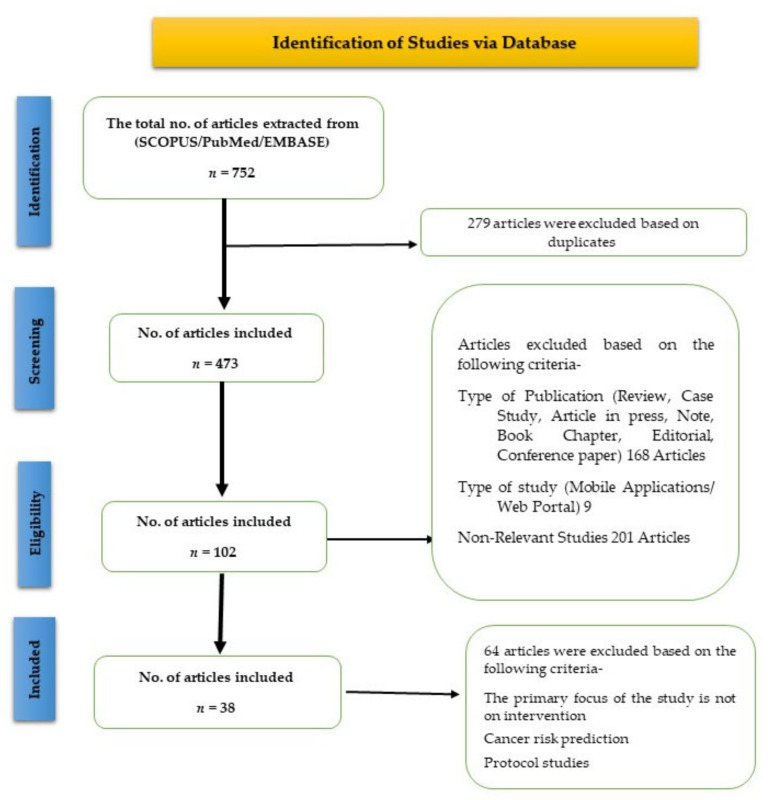
PRISMA flow diagram of the screening and selection of studies.

**Figure 2 cancers-14-04437-f002:**
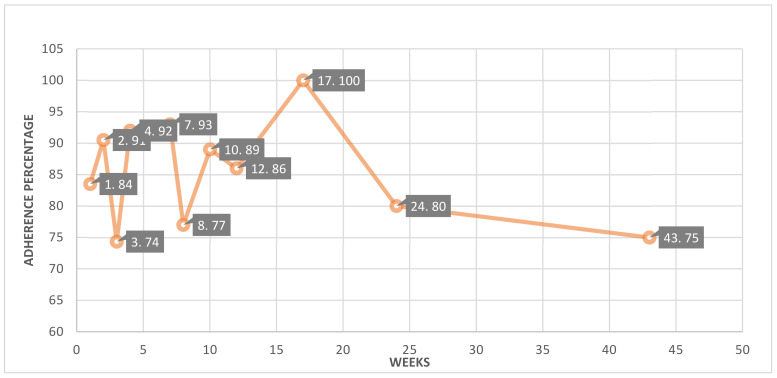
Graphical representation of adherence to wearable devices by duration of intervention.

**Table 1 cancers-14-04437-t001:** Summary of Included studies.

Country of Study	Topic (Type of Cancer and Status)		Study Design	Tools Used	Participant (%) Gender
The United States of America [21]	Breast Cancer (*n* = 57)	Survivors	Feasibility Study	Wearable Device and Questionnaire	100% Women
Australia [22]	Breast Cancer (*n* = 80)	Survivors	RCT	Wearable Device + Text Messages and Personal Interviews + Mobile Application	100% Women
The United States of America [23]	Breast Cancer (*n* = 34)	Survivors	RCT	Wearable Device and Questionnaire (Correlation)	100% Women
Australia [24]	Breast Cancer (*n* = 80)	Survivors	RCT	Wearable Device and Questionnaire (Correlation)	100% Women
The United States of America [25]	Breast Cancer (*n* = 20)	Survivors	RCT	Wearable Device + Group Sessions and Phone Calls	100% Women
Canada [26]	Breast Cancer (*n* = 41)	Survivors	RCT	Wearable Device and Questionnaires (Correlation)	100% Women
The Netherlands [27]	Breast Cancer (*n* = 8)	Survivors	Qualitative Study	Wearable Device and Questionnaires	100% Women
United Kingdom [28]	Breast Cancer (*n* = 39)	Under Treatment	Non-RCT	Wearable Device, Questionnaire, and Behavioral Counseling Session	100% Women
India [29]	Breast Cancer (*n* = 44)	Under Treatment	Non-RCT	Wearable Device + General group session + Questionnaire + Mobile Application	95.4% Women
The United States of America [30]	Breast Cancer (*n* = 32)	Under Treatment	Pilot Study	Wearable Device + Mobile application + Text Messages	100% Women
The United States of America [31]	Breast Cancer (*n* = 10)	Under Treatment	RCT	Wearable Device and Questionnaire	100% Women
Germany [32]	Breast Cancer (*n* = 99)	Under Treatment	Feasibility Study	Wearable Device and Questionnaire	100% Women
Central China [33]	Mixed Cancer (*n* = 112)	Under Treatment	RCT	Wearable Device	76.2% Women
The United States of America [34]	Mixed Cancer (*n* = 38)	Under Treatment	Utility Study/Predictive Study	Wearable Device + Mobile application and Interview	52% Women
The United States of America [35]	Mixed Cancer (*n* = 41)	Under Treatment	Observational Study	Wearable Device + Mobile application + Questionnaire	56% Women
The United States of America [36]	Mixed Cancer (*n* = 33)	Under Treatment	Prospective cohort Study	Wearable Devices and Spirometer	57.5% Women
Japan [37]	Mixed Cancer (*n* = 30)	Under Treatment	Feasibility Study	Wearable Device	70% Men
France [38]	Mixed Cancer (*n* = 31)	Under Treatment	Pilot Study	Wearable Device + Mobile Application + Questionnaire	55% Men
Ireland [39]	Mixed Cancer (*n* = 61)	Survivors	RCT	Wearable Devices + Goal-setting session + Telephone-delivered health-coaching sessions	50% Men
The United States of America [40]	Mixed Cancer (*n* = 32)	Survivors	Feasibility Study	Wearable Device + Two group sessions + support phone call	51% Men
Switzerland [41]	Mixed Cancer (*n* = 30)	Survivors	Feasibility Study	Fitbit + iPad (preloaded apps) + Questionnaires	70% Men
The United States of America [42]	Mixed Cancer (*n* = 59)	Survivors	Pilot Study	Wearable Device and Questionnaire	59.3% Women
The United States of America [43]	Mixed Cancer (*n* = 47)	Survivors	RCT	Wearable Device + Questionnaire + Social Media Intervention (Health Education)	96% Women
Australia [44]	Colorectal and Endometrial cancer (*n* = 29)	Survivors	RCT	Mobile application (in-app chat service) + Wearable device + Questionnaires	58% Women
Western Australia [45]	Colorectal Cancer (*n* = 61)	Survivors	RCT	Wearable Device + mHealth app + Peer-based virtual support group + Qualitative Interviews	50% Women
The United States of America [46]	Colorectal Cancer (*n* = 39)	Survivors	RCT	Wearable Device and Questionnaire-based study	58% Women
South Korea [47]	Colorectal Cancer (*n* = 75)	Under Treatment	Feasibility Study	Wearable device + Questionnaires + e-Patient Diary	58.7% Men
The United States of America [48]	Colorectal Cancer (*n* = 40)	Under Treatment	Pilot Study	Wearable Device and Questionnaire-based study	56.8% Women
Taiwan [49]	Lung Cancer (*n* = 12)	Under Treatment	Observational Study	Wearable Device and Questionnaire-based study	58.33% Men
The United States of America [50]	Lung Cancer (*n* = 30)	Under Treatment	Observational Study	Wearable Device + Questionnaire + Educational handbook + Social support + Email-based coaching	67% Men
The United States of America [51]	Lung Cancer (*n* = 18)	Under Treatment	Observational Study	Wearable Device and Questionnaire (Correlation)	44% Women
South Korea [52]	Lung Cancer (*n* = 555)	Under Treatment	Usability Study	Wearable Devices + Questionnaire+ Educational handbook+ Social support + Email-based coaching	61% Men
The United States of America [53]	Gastric cancer (*n* = 27)	Under Treatment	Cohort Study	Wearable Device + Mobile Application	62.96% Men
Taiwan [54]	Gastric Cancer (*n* = 43)	Under Treatment	Group Study	Wearable Devices + Questionnaires	51% Men
South Korea [55]	Liver Cancer (*n* = 31)	Under Treatment	Usability Study	Wearable Device + Daily text messages+ Questionnaire	84% Men
The United States of America [56]	Blood Cancer (*n* = 11)	Under Treatment	Feasibility Study	Diary + Accelerometer	66.6% Men
Japan [57]	Urothelial Carcinoma (*n* = 21)	Under Treatment	Cohort Study	Wearable Device	84% Men
The United States of America [58]	Skin Cancer (*n* = 60)	Survivor	Observational Study	Wearable Devices + Questionnaire + Interviews	60% Women

**Table 2 cancers-14-04437-t002:** Adherence to wearable devices in the cancer population.

Country of Study	Total Study Duration (in Weeks)	Intervention Duration (in Weeks)	Patients Recruited	Criteria for Evaluation	Adherence (in Percentage)
The United States of America [21]	24	12	60	Percentage of enrolled patients who completed all assessments (10 h per day for 4 days in a week)	95
Australia [22]	24	12	83	Based on the given assessment completion	94
The United States of America [23]	52	24	44	Collection of data (days with less than 1000 steps considered as non-adherent)	65
Australia [24]	12	12	83	Completeness of data collection	96
The United States of America [25]	10	10	30	Completeness of data collection	67
Canada [26]	24	12	45	Completeness of data collection	88
The Netherlands [27]	12	12	10	Based on data collection and total wearing days	80
United Kingdom [28]	2	2	56	Collected data on the different days (39 patients*14 days)	89
India [29]	7	7	44	Users’ tolerance ability to the intensity of the program that was set using the rate of perceived exertion (RPE)	93
The United States of America [30]	17	17	32	Days were considered “valid” if there was any wear time recorded (5 min threshold)	100
The United States of America [31]	10	10	10	Completeness of data collection	100
Germany [32]	24	24	112	Completeness of data collection	95
Central China [33]	8	8	143	Completeness of data collection	78
The United States of America [34]	8	8	45	Completeness of data collection	84
The United States of America [35]	20	8	34	Completeness of data collection	68
The United States of America [36]	43	43	44	Completeness of data collection	75
Japan [37]	4	4	30	Completeness of data collection	90
France [38]	4	4	30	Completeness of data collection	86
Ireland [39]	24	12	68	Completeness of data collection	89
The United States of America [40]	52	12	49	Completeness of data collection	65
Switzerland [41]	12	12	30	Completeness of data collection and qualitative analysis of interviews	83
The United States of America [42]	10	10	59	Completeness of data collection	100
The United States of America [43]	12	12	50	Completeness of data collection	94
Australia [44]	24	12	34	Based on participants who completed the study criterion, which is a minimum of 1000 steps or more denoted per day.	82
Western Australia [45]	12	12	68	Completeness of data collection	94
The United States of America [46]	12	12	41	Completeness of data collection	81
South Korea [47]	12	12	102	Completeness of data collection	74
The United States of America [48]	12	12	44	Completeness of data collection	88
Taiwan [49]	1	1	12	Completeness of data collection	100
The United States of America [50]	1	1	39	Completeness of data collection	67
The United States of America [51]	3	3	30	Completeness of data collection	60
South Korea [52]	52	52	555	Completeness of data collection	100
The United States of America [53]	3	3	41	Based on the rate of data collected during chemotherapy	63
Taiwan [54]	4	4	43	Completeness of data collection	100
South Korea [55]	12	12	37	Equivalent to the completion of the exercise program	84
The United States of America [56]	2	2	12	Completeness of data collection	92
Japan [57]	12	12	28	Completeness of data collection	75
The United States of America [58]	3	3	60	In-person interviews to examine the acceptability of the device and analysis of qualitative data	100

**Table 3 cancers-14-04437-t003:** Reported clinical outcomes in the cancer population using wearable devices.

Country of Study	Cancer Type	Purpose	Reported Clinical Outcomes
The United States of America [21]	Breast Cancer	Behavioral health management (PA/QoL and fatigue)	High engagement among hospitalized patients and increased energy expenditure among cancer survivors. Outcomes depend on numerous factors related to users and their needs.
Australia [22]	Breast Cancer	Behavioral health management (sleep quality)	Changes in actigraphy (sleep efficiency) and PSQI global and subscales favored the intervention arm. Findings were not significant or clinically meaningful.
The United States of America [23]	Breast Cancer	Behavioral health management (physical activity/BMI/QoL/fatigue/ fitness/self-regulation and self-efficacy related to PA)	Self-monitoring, goal setting, and self-efficacy were significantly correlated with activity levels. Increased improvement in health was noted with an increase in PA.
Australia [24]	Breast Cancer	Behavioral health management (MVPA/Sedentary Behavior)	The intervention resulted in increases in MVPA and MVPA accrued in bouts of at least 10 consecutive min while reducing total and prolonged sitting times. A significant difference in MVPA was noted between groups at T2, favoring the intervention arm.
The United States of America [25]	Breast Cancer	Behavioral health management (PA- MVPA, Sedentary/physiological/ psychosocial/QoL variables)	No significant group differences were observed for changes over time for any variable. Both groups showed increased mean daily MVPA, light PA, energy expenditure, and steps/day.
Canada [26]	Breast Cancer	Behavioral health management (PA-MVPA, LIPA, Sedentary Behavior/Sleep quality/health-related Fitness Markers)	Increases in moderate-to-vigorous intensity PA and decreases in sedentary time were significantly greater in the lower-intensity PA group versus the control group at 12 weeks. Increases in V˙O2 max at 12 weeks in both intervention groups were significantly greater than the changes in the control group. Changes in PA and V˙O2 max remained at 24 weeks but differences between the intervention and control groups were not significant.
The Netherlands [27]	Breast Cancer	Behavioral health management (PA-Sedentary behavior)	The activity tracker motivated women to be physically active and increased their awareness of their sedentary lifestyle. Wearing an activity tracker raised lifestyle awareness in patients with breast cancer.
United Kingdom [28]	Breast Cancer	Behavioral health management (Upper Limb Function)	WAM improved on the surgical side of the upper limb with an increment in PA for the first week and showed a good correlation with DASH (0.0506)
India [29]	Breast Cancer	Behavioral health management (Fatigue/QoL//Functional Capacity/PA/Body Composition)	At the end of the 7-week intervention, functional capacity, quality of life, and skeletal mass were significantly improved, whereas fatigue and changes in total fat improved nonsignificantly.
The United States of America [30]	Breast Cancer	Behavioral health management (PA/MVPA/SB/Cognitive functions)	Participants decreased their activity from pre- to post-chemotherapy by 1 h/week in MVPA and 8 h/week in TPA during the decline. This is useful for determining the stage of chemotherapy in which PA starts to decline and patients need extra support for their care.
The United States of America [31]	Breast Cancer	Behavioral health management (PA/Sleep Metrics)	Overall step count decreased by an average of 54 steps per day from baseline during treatment. Although differences in step count, calories expended, and miles walked throughout the RT were minimal, they were significant because of the substantial number of events
Germany [32]	Breast Cancer	Behavioral health management (PA)	Coherence between self-reported and device data was strong (r = 0.566). Neither treatment nor week nor their interaction had effects on step counts. Self-reported activity time was lower for patients receiving chemotherapy than for those not receiving chemotherapy and lower in the 18th week than in the 3rd week
Central China [33]	Mixed Cancer	Behavioral health management (Asleep + QoL)	The baseline measurement was not significantly different among the three groups. However, after the intervention, a significant difference between the experimental and control groups was noted. Sleep quality and PA improved significantly but not the secondary outcomes.
The United States of America [34]	Mixed Cancer	Behavioral health management (Unplanned Healthcare Encounter/PA)	Kinematic features associated with physical activity showed a positive correlation. Chair-to-table kinematics are good predictors of unexpected hospitalization. Get- up-and-walk kinematics are good predictors of low physical activity
The United States of America [35]	Mixed Cancer	Behavioral health management (Unplanned Healthcare Encounter/PA)	This study demonstrated the feasibility of an outpatient wearable activity tracker. The results revealed a 50% disagreement with no association of these disagreements with UHEs and no correlation between the UHEs and ECOG scores. A correlation between (1) average METs and UHEs and (2) no sedentary physical activity hours and UHEs was noted
The United States of America [36]	Mixed Cancer	Behavioral health management (PA/QoL)	Significant improvements across all eight dimensions of HRQOL; most patients (85%) reported that they enjoyed wearing the Fitbit. Most felt that the Fitbit helped them to be more active (79%), whereas a minority (18%) felt their activity level was the same, and none reported becoming less active.
Japan [37]	Mixed Cancer	Behavioral health management (PA/Symptom Burden Assessment/Sleep/Fatigue)	Use of a wearable activity tracker for collecting PGHD in real time according to the protocol was feasible. With respect to adherence, the result was significant. The correlation between the assessed data was not significant
France [38]	Mixed Cancer	Behavioral health management (PA/Sleep)	Results provide evidence for both the feasibility and relevance of the combined objective and subjective remote monitoring of sleep and other symptoms in patients with cancer with single-night precision. This dynamic approach can help the development of novel therapeutics whose testing is warranted in patients with cancer
Ireland [39]	Mixed Cancer	Behavioral health management (MVPA/Cardiovascular risk factors and sedentary behavior)	The estimated difference between groups at 24 weeks supported higher MVPA; no change in MVPA in the intervention group was observed during the 12-week follow-up period, indicating a positive correlation with the improvement in cardiovascular risk factors.
The United States of America [40]	Mixed Cancer	Behavioral health Management (MVPA/QoL/ Fatigue/Fitness/Sedentary Behavior)	Results of the studies revealed some promising improvements in muscular strength that aligned with the intervention’s focus on strength training.
Switzerland [41]	Mixed Cancer	Behavioral health management (Symptom Analysis)	Remote monitoring of healthcare status in patients receiving palliative care with a limited life expectancy is feasible, and patients can handle the smartphone and sensor-equipped bracelet. Feedback toward the use of this monitoring system was mostly positive.
The United States of America [42]	Mixed Cancer	Behavioral health management (PA-SB and MVPA/QoL)	Intervention participants had a lower-than-expected engagement in the Facebook group component, (passive instead of active engagement); MVPA and sedentary time showed no significant difference b/w gaps
The United States of America [43]	Mixed Cancer	Behavioral health management (PA-MVPA)	Increased physical activity among cancer survivors was noted: the intervention group increased their daily steps. Moderate-to-vigorous-intensity activity performed in 10 min bouts increased, but no significant group-by-time differences for either light- or vigorous-intensity activity were noted
Australia [44]	Colorectal and endometrial Cancer	Behavioral health management (PA -Steensma)	Fitbit wear time (percentage of valid wear days = adherence) was consistent with a median adherence score of 100%. Comparison and correlation with actigraphy (MVPA) show that both devices are not correlated and do not show any type of association.
Western Australia [45]	Colorectal Cancer	Behavioral health management (MVPA/Cardiovascular Risk)	Despite a significant increase in MVPA, the change in the proportion of participants meeting the guidelines in relation to MV10 did not significantly differ by group. Reduction in DBP among intervention participants that were hypertensive. Fitbit was promising for low-intensity interventions.
The United States of America [46]	Colorectal Cancer	Behavioral health management (PA-MVPA/ Adverse events)	Intervention arm increased its MVPA by 13 min per day more than the control arm. Larger studies should be conducted to determine whether the intervention increases physical activity.
South Korea [47]	Colorectal Cancer	Behavioral health management (PA/QoL/Nutritional Status/Physical Performance)	Lower-extremity strength and cardiorespiratory endurance were significantly improved. Fatigue and nausea/vomiting symptoms were significantly relieved after the program. Most of the functional scales showed improvements, although the changes were not significant.
The United States of America [48]	Colorectal Cancer	Behavioral health management (PA/)	Pilot data show a nonsignificant decrease in moderate activity accumulated in bouts of at least 10 min in both arms (16–21 min per week).
Taiwan [49]	Lung Cancer	Behavioral health management (CRF)	The LF to HF ratio is highly correlated with the subjective BFI, particularly when measured during sleep time. Analytical results revealed that this ratio can be used to evaluate cancer fatigue because of a 3% mapping error in the BFI
The United States of America [50]	Lung Cancer	Behavioral health management (Steps/Day and MVPA/Sedentary Behavior/Cardiorespiratory Fitness)	Participants who received surgery in the spring, summer, autumn, and winter seasons, respectively, had lower PA and CRF than those who received surgery in other seasons. These results were consistent among all study subgroups.
The United States of America [51]	Lung Cancer	Behavioral health management (PA-Steps/QoL/ Symptoms/Functional Status/Dyspepsia)	Improved PA was associated with the early discharge of patients with GC undergoing gastrectomy. This was because patients with improved PA had resumed physical function, which was the main factor evaluated if patients were qualified to be discharged.
South Korea [52]	Lung Cancer	Behavioral health management (MVPA/Aerobic Capacity)	Eight (47%) of the seventeen participants demonstrated a clinically significant improvement of 14 m or more. The average improvement in aerobic capacity (13.8 m) was close to the minimum threshold for a clinically meaningful improvement of 14 m
The United States of America [53]	Gastric cancer	Behavioral health management (PA and Symptom Burden)	This study’s results indicate significant correlations between the number of the step count and two common performance statuses, which is consistent with previous research findings. Questionnaire findings indicated that active patients have a lower burden of symptoms.
Taiwan [54]	Gastric Cancer	Behavioral health management (PA/Sleep Metrics)	Results provide evidence for both the feasibility and relevance of the combined objective and subjective remote monitoring of sleep and other symptoms in patients with cancer with single-night precision. This dynamic approach can guide the development of novel therapeutic concepts whose testing is warranted in patients with cancer
South Korea [55]	Liver Cancer	Behavioral health management (Exercise Capacity/PA/QoL/Body Composition and Biochemical)	Compared with baseline, significant improvements were found in physical fitness measures, body composition, self-reported amount of physical activity, and pain. All symptoms improved, as observed in the QoL scales (i.e., EORTC-QLQ C30).
The United States of America [56]	Blood Cancer	Behavioral health management (PA/Sleep)	This study demonstrates the feasibility of collecting sleep data through actigraphy among hospitalized adults. Actigraphy measures suggested poor sleep.
Japan [57]	Urothelial Carcinoma	Behavioral health management (PA/QoL/ Adverse Events)	Significant correlations were noted between measurements performed using an oscillometer and a Fitbit during chemotherapy for patients. The measurement of fatigue using Fitbit was effective
The United States of America [58]	Skin Cancer	Preventive care	No differences in baseline knowledge or attitudes regarding sun exposure or protection were noted between the two groups.

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
