# Peer review of "A Scoping Review to Assess Adherence to and Clinical Outcomes of Wearable Devices in the Cancer Population"

_cancers, 2022, doi:10.3390/cancers14184437_

Round 1

Reviewer 1 Report

The issue of adherence of wearable devices in cancer patients is of growing relevance, due to the explosive increment of the use of these devices.  The paper is methodologically well explained, and follows the international guidelines. However, there a some points that require further clarification:

1. One major point is the fact recognized by th eauthors that the definition of adherence follows the one defined by each paper. Then, it seems feasible that the authors are comparing adherence data which are not strictly comparable. This shoul dbe mentioned as a limitation. I think that the paper would be improved significantly if the authors explain the different definitions found in a table.

2. There is no option to do a meta-analysis as it could be assumed by the reader, but I think it whoucl be explicitely mentioned. 

3. It is not clear if there is any statistical analysis carried out. No description in the methodlogy but in the results SPSS is mentioned when assessing adherence trends. Please, clarify.

4. There is incosnistency in the wording of adherence. In the result section 3.7.1 two times 'compliance' is mentioned instead of adherence, whihc is a mistake. I think the terminology should be consistent.

5. Section 3.8. on clinical outcomes: the variables included in this section are highly variable and difficult to compare. I would use a more general word instead of clinical outmes (one wait for survival or disease free survival, and we found process variables). Also, the issue of different criteria and variables used should be mentioned

6. Limitations of the study should be expanded also to limitations o fthe studies. There is not any attempt of the authors to assess the quality of the sstudies reviewed. This is a major problem because the paper is then purely descriptive, without any attempt to assess what the authours found. I think this is the major problem of the paper.

Minor problesm that require review for clarification:

- abstract line 35-36: Which correlation is done? Not reported neither in the result section, nor in the methodology. PLease, clarify

- lines 47 to 52. Perhaps better after definitions. Also, a survey is mentioned, which survey? which territory?

-lines 71-76: please review for comprehension. I think this sentence is not clear.

- lines 141-3. It should be in the following section. By the waty, which analysis are carried out? Not expalined at all. 

In summary, interesting paper but purely descriptive without any assessment of the quality of the papers under review. This should be included as limitation. The analysis of the adherence definition would be very interesting, as well as the outcomes of the studies. The paper could be much improved with a better analytical approach.

Author Response

We are extremely thankful to the reviewer for his/her valuable comments and suggestions that have inspired several changes in our work and we hope to have significantly improved it. Please find below, the point-to-point response to the reviewer’s suggestions.

Point 1: One major point is the fact recognized by the authors that the definition of adherence follows the one defined by each paper. Then, it seems feasible that the authors are comparing adherence data that are not strictly comparable. This should be mentioned as a limitation. I think that the paper would be improved significantly if the authors explain the different definitions found in a table.

Response 1: Thank you for your feedback on this research. We agree with your synthesis.

Following your suggestions, we have added a few more lines that define adherence clearly in the methodology (lines 139-140). In addition to this, we have modified table 2 and specified the definition of adherence for each paper. In the limitations section, we have mentioned these points in accordance with your suggestion.

Point 2: There is no option to do a meta-analysis as it could be assumed by the reader, but I think it would be explicitly mentioned. 

Response 2: Thank you, we agree with your suggestion and provided explanations in the limitation section as follows-

“We were unable to conduct a meta-analysis or systematic review due to numerous variations in study design, clinical outcome, and adherence definitions. As a result, we conducted a scoping review and presented a tabulated analysis of studies.”

Point 3: It is not clear if there is any statistical analysis carried out. No description in the methodology but in the results, SPSS is mentioned when assessing adherence trends. Please, clarify.

Response 3: Thank you for pointing it out. We have now clarified this point in our methodology (in lines 143-144).

Point 4: There is inconsistency in the wording of adherence. In the result section 3.7.1 two times 'compliance' is mentioned instead of adherence, which is a mistake. I think the terminology should be consistent.

Response 4: Thank you for bringing it to our attention. We have corrected this mistake in the result section 3.7.1.

Point 5: Section 3.8. on clinical outcomes: the variables included in this section are highly variable and difficult to compare. I would use a more general word instead of clinical outcomes (one wait for survival or disease-free survival, and we found process variables). Also, the issue of different criteria and variables used should be mentioned.

Response 5: Thank you for your comments on this study. We agree with the suggestions and provided an additional explanation as follows-

The studies included in our review are majorly focused on behavioral health management which includes different clinical outcomes such as physical activity, sleep quality, quality of life, unplanned health-care encounters, MVPA, sedentary behavior, and symptom burden. Every paper has its own definition of clinical outcomes among cancer survivors and cancer patients undergoing treatment. For Example-

  1. In one paper [26], the wearable device motivated cancer survivors to be physically active.
  2. In one paper [33], they focused on unplanned healthcare encounters and physical activity.
  3. In another paper [37], they focused on the objective and subjective remote monitoring of sleep and other symptoms in patients with cancer.

We have covered all of this data regarding clinical outcomes in table 3. Besides this, we have noted the issue of different criteria (purposes) and variables (reported clinical outcomes) in our limitations. 

Point 6: The limitations of the study should be expanded also to limitations of the studies. There is not any attempt by the authors to assess the quality of the studies reviewed. This is a major problem because the paper is then purely descriptive, without any attempt to assess what the authors found. I think this is the major problem of the paper

Response 6: We appreciate your feedback. Though we did quality assessment in terms of adherence and clinical outcomes. Since we can’t perform a meta-analysis, we defined and reported adherence as well as clinical outcomes in Tables 2 and 3. We have also added the following statement in the limitation- 

“We were unable to conduct a meta-analysis or systematic review due to numerous variations in study design, clinical outcome, and adherence definitions. As a result, we conducted a scoping review and presented a tabulated analysis of the studies”.

Minor problems are addressed as follows:

Comment 1: Abstract lines 35-36: Which correlation is done? Not reported neither in the result section nor in the methodology. Please, clarify

Response 1: Thank you for your comment. In our scoping review, 10 out of 38 studies reported a correlation between the subjective questionnaires (patients reported outcomes) and wearable outcomes. We have mentioned these outcomes in the results section 3.8 as follows-

 “Most of the studies in this review evaluated the feasibility of and adherence to WDs. Of the 38 studies, only 10 determined the correlation between patient-reported and objective outcomes; these studies reported that the use of WDs significantly improved clinical outcomes (27,31,34,36,37,48,49,52,54,56).”

Comment 2: lines 47 to 52. Perhaps better after definitions. Also, a survey is mentioned, which survey? which territory?

Response 2: Thank you for pointing it out. The survey mentioned in lines 47-52 earlier was not conducted by Deloitte global. Hence, it has been removed from the paper.

Comment 3: lines 71-76: Please review for comprehension. I think this sentence is not clear.

Response 3: Thank you for your suggestion. We have rephrased the sentences in order to make them more comprehensive.

Comment 4: - lines 141-3. It should be in the following section. By the way, which analyses are carried out? Not explained at all. 

Response 4: Thank you for pointing this out. We have modified the lines and defined the analysis in a more effective way

Reviewer 2 Report

Journal: Cancers by MDPI

Title: A Scoping Review to Assess Adherence to and Clinical Outcomes of Wearable Devices in the Cancer Population

Comments for Authors

        This review focuses on wearable devices in medical applications and cancer therapy. A systematic methodology was applied to screen the review articles from database and summarize in terms of adherence and outcomes. This review paper is comprehensive and informative but would need a major revision, particularly for the adherence study which is the major issue. Please check the following comments.

1.   Some paragraphs, mostly in Introduction, have inconsistent fonts (Palatino and Times Roman) and sizes. Need to unify.

2.     Reference for the 2nd cause of death should be added. (Line 74)

3.   The cross-reference of Table 3 in Line 136 should be Table 1. Tables should be numbered as the order of their presence.

4.     The caption of Figure 1 should be added.

5.     What is the last column consisting numbers in Table 1?

6.     Several problems in 3.7.1., Table 2, and Figure 2.

          i.              The second paragraph is very confusing. Need to be polished.

         ii.              The increase of adherence rate vs. intervention period in Figure 2 is not obvious. Without standard errors and correlation study, it is hard to conclude the claim. Moreover, the adherence data decreased after a certain intervention time in Figure 2 doesn’t correspond to the conclusion.

       iii.              Which study is with a duration of 52 weeks? Need a direct reference.

       iv.              Figure 2 is plotted from intervention periods instead of follow-up periods, so one cannot conclude that the adherence rate of WD decreases as follow-up time increases because no follow-up data was shown here. Frankly, Figure 2 is quite puzzling and certainly needs to be redesigned to another style like correlation or quadrants. In addition, it doesn’t reflect the number of studies with certain intervention weeks, making the adherence data hard to understand (there are 38 studies but only 11 data points).

         v.              There are only 5 studies with 12 weeks of intervention and 12 weeks of follow-up. Ref 31 has no follow-up period according to Table 2. Also, it is difficult to find the follow-up periods in Table 2 (no definition there, I assumed that it is the time of total period subtracted by the intervention period).

       vi.              In Table 2, the fonts of “Patients Recruited” and “Adherence” are different. Please make sure the consistency in the entire article.

      vii.              A misused format of 1200% in Table 2.

7.   In 4.2., “Changes in study parameters” is not clear. What changes? What parameters?

8.    Some minor grammar errors in Conclusion: 1) Adherence to the wearable devices gets “affected” 2) Sentence structure: There is no verb in the sentence of “The better understanding…much needed.”

Author Response

Response to Reviewer 2

We are extremely thankful to the reviewer for his/her valuable comments and suggestions that have inspired several changes in our work and we hope to have significantly improved it.  Please find below, the point-to-point response to the reviewer’s suggestions.

Point 1 Some paragraphs, mostly in Introduction, have inconsistent fonts (Palatino and Times Roman) and sizes. Need to unify.

Response 1: Thank you for pointing it out. We have made the appropriate changes in the introduction.

Point 2:     Reference for the 2nd cause of death should be added. (Line 74)

Response 2: Thank you for your feedback on this research. We have added the required reference to the text in line 74.

Point 3: The cross-reference of Table 3 in Line 136 should be Table 1. Tables should be numbered in the order of their presence.

Response 3: Thank you for your suggestion. We have put tables in the right order of sequence. However, in line 136, we have removed the whole sentence because it was not analyzed in the current review.

Point 4: The caption of Figure 1 should be added.

Response 4: Thank you for bringing this to our attention. We have added the required caption for figure 1 

Point 5: What is the last column consisting of numbers in Table 1?

Response 5: Thank you for pointing this out. The last column signified the intervention period for each study. However, we have deleted this column because it is already mentioned in table 2. 

Point 6: Several problems in 3.7.1., Table 2, and Figure 2

Comment 1: The second paragraph is very confusing. Need to be polished.

Response 1: Thank you for your input. We have edited the second paragraph of section 3.7.1 and tried to explain it more clearly. 

Comment 2: The increase in adherence rate vs. intervention period in Figure 2 is not obvious. Without standard errors and correlation studies, it is hard to conclude the claim. Moreover, the adherence data decreased after a certain intervention time in Figure 2 doesn’t correspond to the conclusion.

Response 2:  Thank you for the suggestion. We have edited our results section as per the newly modified figure 2.

Comment 3: Which study is with a duration of 52 weeks? Need a direct reference.

Response 3: Thank you for bringing this to our attention. We have added the required reference for this study

Comment 4: Figure 2 is plotted from intervention periods instead of follow-up periods, so one cannot conclude that the adherence rate of WD decreases as follow-up time increases because no follow-up data was shown here. Frankly, Figure 2 is quite puzzling and certainly needs to be redesigned to another style like correlation or quadrants. In addition, it doesn’t reflect the number of studies with certain intervention weeks, making the adherence data hard to understand (there are 38 studies but only 11 data points).

Response 4: Thank you for your suggestion. The focus of our study was the intervention duration, not the follow-up period. Therefore, the average adherence rate for only intervention duration has been plotted in figure 2.  The figure has been slightly modified. 11 data points are basically 11 segments “based on the duration of each study's intervention. The average adherence percentage was calculated for each segment. The intervention-based segments included studies with weeks 1 (48,49), 2 (27,55), 3 (50,52,57), 4 (36,37,53), 7(28), 8 (32,33,34), 10 (24,30,41), 12 (20,21,23,25,26,38,39,40,42,43,44,45,46,47,54,56), 17 (29), 24 (22,31,), and 43 (35)”. We have added these lines in the second paragraph of results section 3.7.1.

Comment 5: There are only 5 studies with 12 weeks of intervention and 12 weeks of follow-up. Ref 31 has no follow-up period according to Table 2. Also, it is difficult to find the follow-up periods in Table 2 (no definition there, I assumed that it is the time of the total period subtracted by the intervention period).

Response 5 Thank you for pointing this out. We have made the appropriate changes in the text according to table 2. You assumed correctly that the follow-up period was calculated by subtracting intervention duration from the total study duration. We have also added this sentence to results section 3.7.1.

Comment 6: In Table 2, the fonts of “Patients Recruited” and “Adherence” are different. Please make sure of consistency in the entire article.

Response 6 Thank you for your suggestion. We have made the appropriate changes in the whole document

Comment 7: A misused format of 1200% in Table 2.

Response 7: Thank you for bringing this to our attention. We have corrected the information in the table.

Point 7: In 4.2., “Changes in study parameters” is not clear. What changes? What parameters?

Response 7 Thank you for your comment. By study parameters, we meant to highlight the existence of variations in the adherence criteria for all the included studies. We removed this sentence because it had already been explained in the preceding sentences.

Point 8: Some minor grammar errors in Conclusion: 1) Adherence to the wearable devices gets

“affected” 2) Sentence structure: There is no verb in the sentence of “The better understanding…much needed.”

Response 8: Thank you for pointing it out. We have done proof reading and made the required changes.

Round 2

Reviewer 1 Report

thanks for the detailed answers to my comments and points raised. I have no further comments 

Reviewer 2 Report

Journal: Cancers by MDPI

Title: A Scoping Review to Assess Adherence to and Clinical Outcomes of Wearable Devices in the Cancer Population

Comments for Authors

        The authors address most of the comments in the previous round of review. The remaining problems are formatting-related, so a minor revision is recommended. Please thoroughly proofread and pay attention to any inconsistency and formatting issues.

1.     Same issue “Tables should be numbered as the order of their presence." The “Table 2” mentioned in Line 139 is actually the first cross-reference of tables, meaning that it should be Table 1.

2.     The “coordinates” in Figure 2 need to be adjusted because some of them are covering the data points and trend. The visualization needs to be improved.

3.     Please unify the fonts in all tables. Especially in Figure 2, the title, the columns of “Patients Recruited” and “Adherence”, and the context have three different fonts. This issue in the previous comment hasn’t been resolved at all. It is of the authors responsibility to ensure the consistency.

4.     Different fonts again in reference. Also, through the entire article, several misaligned paragraphs, spacing, and session titles as well as misused text bolding (Line 309-317) and double periods were found. Please have the manuscript carefully proofread before submission to fix any formatting issues and take this seriously as inconsistency significantly distracts readers and degrades the quality of an article.
